# Involvement of community pharmacy pharmacists in fecal immunochemical test screening without government support in Japan

Yusuke Jin'o[1,2], Miharu Ushikai[3], Yuga Komaki[3], Koichi Masuda[3], Masahisa Horiuchi[3]*

**1** Momiji Pharmacy, Yoshishige Pharmacy Group, Kagoshima, Japan, **2** Kagoshima City Pharmaceutical Association, Kagoshima, Japan, **3** Department of Hygiene and Health Promotion Medicine, Kagoshima University Graduate School of Medicine and Dental Sciences, Kagoshima, Japan

* masakun@m.kufm.kagoshima-u.ac.jp

## Abstract

Colorectal cancer (CRC) is the third most common malignancy and second leading cause of death worldwide. However, the screening rate, which is a typical preventive measure, remains low. A community pharmacy pharmacist (CPP)-mediated procedure was used to increase the CRC screening rate. A total of 37 community pharmacies in Kagoshima, a core city in Japan, participated in this study. The results were statistically compared with the results of two procedures in Kagoshima City: hospital/clinic institution-mediated and health examination institution-mediated procedures. The cost was set at 1,100 JPY to perform a fecal immunochemical test, considering the costs as the self-payment of the other two procedures. In March 2023, 2,611 kits were distributed, and 273 tests were conducted under the research conditions. A significantly higher percentage of people in their 40s were tested using CPP-mediated procedures (35.2% vs. 14.3% in the hospital/clinic, and 21.8% in the health examination institution, respectively, $p < 0.01$). The percentage of participants who underwent a detailed examination at CPP facilities was significantly lower than in the other two groups (46.7% vs. 87.3% in the hospital/clinic, and 83.0% in the health examination institution, $p < 0.05$). In the CPP-mediated procedure, the test result turn-around time was approximately one day. Although a 2-day method was used, the implementation rate was 100%. The CPP-mediated procedure did not differ significantly from hospital/clinic or health examination institution procedures in terms of the number of detailed examination findings. In particular, the CPP-mediated procedure may address gaps by increasing the number of young people and people with reduced opportunities who receive CRC screening. The CPP-mediated procedure could be implemented as a new procedure with certain advantages. Moreover, it should be considered that this procedure can be implemented and sustained in society without government support.

**Data availability statement:** Data on the number of pharmacists in the pharmacies were obtained from the site (https://iryo-info.pref.kagoshima.jp/qqport/), which was updated on December 19, 2023 (Table 1).

Data on participants of hospital/clinic and health examination institutions for CRC screening in Kagoshima city in 2018 were obtained from the website (https://www.e-stat.go.jp/stat-search/files?page=1&layout=datalist&toukei=00450025&t-stat=000001030884&cycle=8&t-class1=000001155266&t-class2=000001155275&t-class3val=0&metadata=1&data=1) before the COVID-19 pandemic in Japan (Tables 2 and 3).

**Funding:** The author(s) received no specific funding for this work.

**Competing interests:** The authors have declared that no competing interests exist.

## Introduction

Colorectal cancer (CRC) is the third most common malignancy and second leading cause of death globally, and a public health problem in Japan [1,2]. Additionally, recently, there has been an increase in the global incidence of CRC in adults aged <50 years [3]. To improve CRC prognosis, there is an urgent need to establish preventive medical procedures and improve treatment outcomes.

Screening in Japan is conducted under the Health Promotion Law, and municipalities are responsible for its implementation through financial support in addition to individual funding. The screening rate of typical preventive procedures, including fecal immunochemical testing (FIT), is low [4,5]. In Japan, there are two FIT procedures that are provided by hospitals/clinics and health examination institutions. The usefulness of screening has been confirmed [6,7]. However, the screening rate for primary screening is not high, and needs improvement [8–10]. Although improvements in the public awareness and promotion of CRC screening by physicians will help raise participation rates, physicians' capacity is limited. Although various efforts have been made to increase CRC screening rates, such as the distribution of CRC testing kits to participants through a health checkup program or at influenza vaccination clinics, screening rates remain unsatisfactory [10,11].

Community pharmacy pharmacists (CPPs) deliver public health services for preventive medical care worldwide [12–14]. These services include promoting smoking cessation, weight management programs, and syringe exchange and inoculation programs. However, CPPs in Japan have not been significantly involved in the provision of health services provided by public facilities, independent of their work [15]. Currently, pharmacists in Japan are undergoing a shift to a six-year system in university, which includes the expansion of the educational content related to prevention and health in addition to clinical training [16]. Along with involvement in medical care by providing medicine, pharmacists' work in supporting self-care using over-the-counter drugs functions as a health support activity by the pharmacy. Moreover, CPPs are expected to be involved in preventive medical initiatives provided by regional healthcare centers or insurers. The use of CPPs has been devised to improve access and knowledge of the CRC test, and has been implemented in various countries, including Italy, Switzerland, and Spain [17–19].

A preliminary study on the possibility of implementing CPP-mediated CRC screening was conducted in Kagoshima, a core city in Japan. This study aims to report the involvement of CPPs in secondary preventive medical initiatives for CRC screening in Japan.

## Materials and methods

### Study design and procedure

This study was performed as a survey and included 37 community pharmacies in Kagoshima, based on voluntary participation (Table 1). The participating pharmacists were trained in a general briefing (supplementary material) before the implementation. Kits were distributed by participating pharmacies between March and the first week of April 2023 (Fig 1). Stool samples were collected at home, brought to a community pharmacy, and sent to the laboratory (Clinical Pathology Laboratory Co. Ltd.,

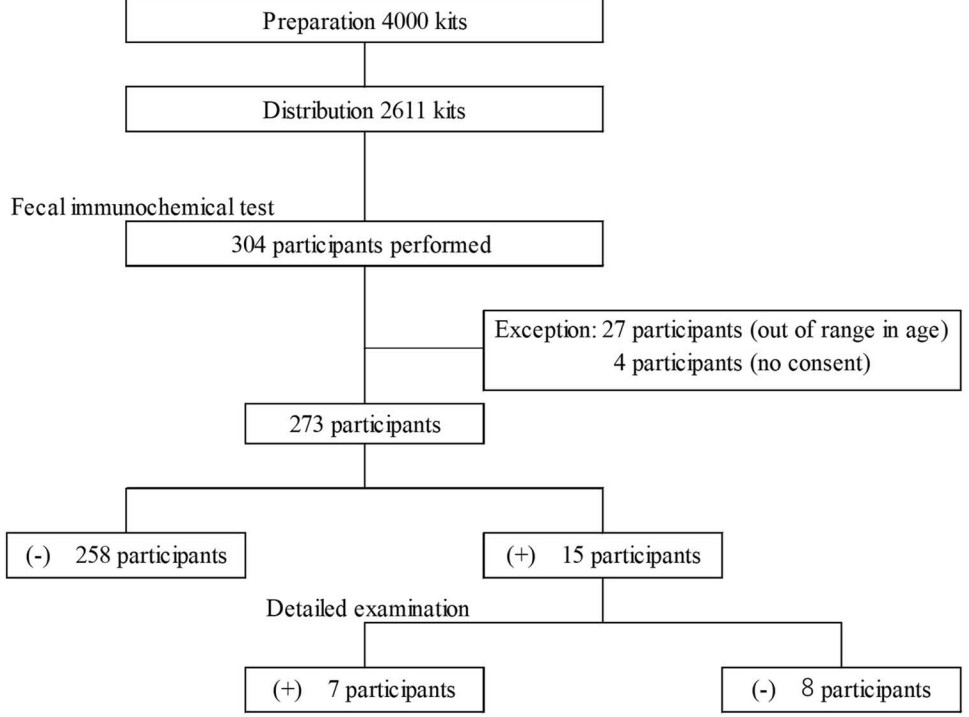

**Fig 1. Flow chart of distributed and collected fecal immunochemical tests at community pharmacies, and the results of the detailed examination.**

Kagoshima, Japan). The collection period was from March to August 31, 2023. The results were reported to the community pharmacist in charge, and then to the participants. For patients who tested positive, explanations were provided about the medical institutions that performed detailed examinations.

The FIT kits were manufactured by Eiken Chemical Co. Ltd. (Tokyo, Japan). Considering the sensitivity of the test, a 2-day method was used, which was the same as the method usually used for CRC screening in Japan [20]. The cut-off value was set at 100 ng Hb/ml as per the manufacturer's recommendations. Therefore, the sensitivity of the 2-day method for CRC is 100.0% (74.1–100.0% as 95% confidential intervals) [20].

A total of 31 participants were excluded either due to the absence of consent to participate in the study or age outside the study range (40–69 years old). Finally, 273 participants who underwent FIT were included in the study (Fig 1). Participant agreed to pay 1100 JPY for FIT, which was the same as the cost at medical institutions for CRC screening in Kagoshima. Comparatively, hospital/clinic mediated- and health examination institution mediated-procedures cost patients 600 JPY and 1100 JPY, respectively, as per the rules of the local government. Furthermore, those who agreed to undergo FIT were asked to complete a questionnaire to investigate their characteristics, including their awareness of health check-ups. All Japanese people must register with health care insurers based on their age and occupational status. Notably, health services, including cancer screening tests, differ for the respective insurers. Therefore, the participants were also asked to name their medical health insurer in the administered questionnaire.

## Data collection method

Data on the number of pharmacists in each pharmacy were obtained from the website https://iryo-info.pref.kagoshima.jp/qqport/, which was updated on December 19, 2023 (Table 1). Data on participants of hospital/clinic and health

**Table 1. Distribution of the kits and the tests examined in the participating pharmacies.**

| Pharmacy | Number of pharmacists[a] | Number of kits distributed | Number of FIT examined | Number of positive results |
|---|---|---|---|---|
| A | 6.0 | 668 | 100 | 1 |
| B | 14.4 | 228 | 14 | 2 |
| C | 4.9 | 176 | 19 | 1 |
| D | 2.0 | 150 | 15 | 0 |
| E | 5.0 | 140 | 12 | 2 |
| F | 4.2 | 126 | 7 | 0 |
| G | 2.5 | 111 | 2 | 0 |
| H | 1.7 | 89 | 4 | 0 |
| I | 1.1 | 75 | 12 | 1 |
| J | 3.0 | 67 | 5 | 2 |
| K | 2.0 | 64 | 10 | 2 |
| L | 3.0 | 61 | 11 | 2 |
| M | 3.8 | 56 | 2 | 0 |
| N | 2.0 | 52 | 1 | 0 |
| O | 3.2 | 50 | 1 | 0 |
| P | 4.0 | 47 | 6 | 0 |
| Q | 1.0 | 45 | 5 | 0 |
| R | 3.0 | 42 | 6 | 0 |
| S | 2.0 | 40 | 2 | 0 |
| T | 2.0# | 36 | 2 | 0 |
| U | 1.8 | 33 | 1 | 0 |
| V | 2.2 | 32 | 4 | 0 |
| W | 2.0 | 30 | 1 | 0 |
| X | 1.0 | 29 | 3 | 0 |
| Y | 1.5 | 26 | 1 | 0 |
| Z | 2.0 | 21 | 2 | 0 |
| AA | 2.9 | 20 | 3 | 0 |
| BB | 1.7 | 20 | 4 | 0 |
| CC | 4.5 | 15 | 7 | 1 |
| DD | 2.5 | 13 | 1 | 0 |
| EE | 2.0 | 11 | 1 | 0 |
| FF | 3.0 | 10 | 3 | 1 |
| GG | 1.0 | 9 | 0 | 0 |
| HH | 6.0 | 7 | 1 | 0 |
| II | 2.0 | 5 | 4 | 0 |
| JJ | 4.0 | 4 | 1 | 0 |
| KK | 2.0 | 3 | 0 | 0 |
| Sum, *n* (%) | | 2611 | 273 (10.5)[b] | 15 (5.5)[c] |

FIT: fecal immunochemical test.

[a]The numbers (except T) were calculated based on the public data at the end of March, 2023.

#The number of pharmacists at pharmacy T was informed personally to the researchers because it was not registered on the data.

[b]The percentage was calculated based on the number of FIT kits examined divided by the number of kits distributed.

[c]The percentage was calculated based on the number of positive FIT results divided by the number of FIT kits examined.

examination institutions for CRC screening in Kagoshima in 2018 were obtained from https://www.e-stat.go.jp/stat-search/files?page=1&layout=datalist&toukei=00450025&tstat=000001030884&cycle=8&tclass1=000001155266&t-class2=000001155275&tclass3val=0&metadata=1&data=1 (Tables 2 and 3). This time period was selected to avoid the impact of the COVID-19 pandemic, which affected screening rates of general health examinations, including CRC screening.

The data demonstrated in Tables 1–3 were first accessed on April 9, 2024, and July 14, 2023, respectively.

## Statistics

Categorical data are expressed as actual numbers. Statistical analyses were performed using the chi-squared test or Fisher's exact test, as appropriate. The chi-square test was followed by residual analysis for comparison between groups. EZR (version 3.4.3) [21] and js-STAR (release 1.9.7 j; https://www.kisnet.or.jp/nappa/software/star/freq/chisq_ixj.htm) were used for statistical analyses when appropriate.

## Ethics

Ethical approval was obtained from the Institutional Review Committee of the Kagoshima University Hospital (approval number: 220199 Eki). Written informed consent was obtained from all the enrolled participants.

## Results

### Participating community pharmacy characteristics

As shown in Table 1, 37 community pharmacies in Kagoshima (353 registered pharmacies as of May 1, 2024) participated in this study. Participation was voluntary. The number of pharmacists working in the participating pharmacies ranged from 1.0 to 14.4. The number of working pharmacists was calculated based on the employment conditions stated in the Pharmaceutical Affairs Law. There was a weak correlation between the number of pharmacists in the pharmacy and the number of kits distributed (r = 0.454, $p < 0.01$). However, no significant difference was observed when comparing the number of kits distributed between pharmacies with two or fewer pharmacists and those with more than two working pharmacists. As a result, 273 individuals were examined in community pharmacies, and 15 positive tests were obtained (Fig 1).

### Participant characteristics

The 273 participants who underwent FIT had health insurance with the following insurers: 166 (60.8%) participants were with the Japan Health Insurance Association, 71 (26.0%) were with the National Health Insurance, 16 (5.9%) were with the Mutual Aid Association, 15 (5.5%) were with Union Health Insurance, 3 (1.1%) were with the Public Fund, 1 (0.4%) was with the Late-Stage Senior Citizen's Health Care System Insurer, and 1 (0.4%) participant's insurer was unknown. A total of 127 (46.5%) participants were undergoing the FIT for the first-time, while 146 (53.5%) participants had been tested at least once previously. The number of patients regularly taking medication was 170 (62.3%), whereas 102 (37.4%) did not receive any medication. Regarding subjective interest in health checkups, 7 (2.6%) participants had low interest, 67 (24.5%) had moderately low interest, 159 (58.2%) had moderately high interest, 39 (14.3%) had high interest, and the interest of 1 (0.4%) participant was unknown. Although the interest rate was relatively high, 27.1% of the participants showed that it was either low or low.

### Comparison between the community pharmacy pharmacist implementation group, hospital/clinic implementation group, and health examination institution implementation group

In Japan, there are two FIT procedures, mediated by hospitals/clinics and health examination institutes. Their data are publicly available from the Ministry of Health, Labor, and Welfare. In Table 2, the current data was compared with the two 2018 data points from Kagoshima, as described above. There were 2,192 examinees at the hospital/clinic and 8,403 at the Health

Table 2. Comparison of the fecal immunochemical blood test results among the three types of the procedure.

| | Procedures | | | A vs B | A vs C | B vs C |
|---|---|---|---|---|---|---|
| | CPP (A) | *Hospital/clinic (B) | *Health examination institution (C) | | | |
| Participants examined, n | 273 | 2192 | 8403 | | | |
| Sex | | | | NS | NS | NS |
| Male, n (%) | 79 (28.9) | 664 (30.3) | 2285 (27.2) | | | |
| Female, n (%) | 194 (71.1) | 1528 (69.7) | 6118 (72.8) | | | |
| Age, years | | | | p<0.01 | p<0.01 | p<0.01 |
| | | | | A B | A C | B C |
| 40-49, n (%) | 96 (35.2) | 313 (14.3) | 1835 (21.8) | △ ▼ | △ ▼ | ▼ △ |
| 50-59, n (%) | 73 (26.7) | 371 (16.9) | 1645 (19.6) | △ ▼ | △ ▼ | ▼ △ |
| 60-69, n (%) | 104 (38.1) | 1508 (68.8) | 4923 (58.6) | ▼ △ | ▼ △ | △ ▼ |
| Experience of FIT | | | | NS | NS | NS |
| Yes, n (%) | 146 (53.5) | 1139 (52.0) | 4776 (56.8) | | | |
| No, n (%) | 127 (46.5) | 1053 (48.0) | 3627 (43.2) | | | |
| Characteristics of the procedure of FIT | | | | | | |
| Completion of two times examination, n (%) | 273 (100) | NA | NA | | | |
| ¹Days for return, days | 1.1±0.3 | NA | NA | | | |
| Positive results | | | | NS | NS | NS |
| Yes, n (%) | 15 (5.5) | 126 (5.7) | 423 (5.0) | | | |
| No, n (%) | 258 (94.5) | 2066 (94.3) | 7980 (95.0) | | | |

CPP: community pharmacy pharmacist, FIT: fecal immunochemical test.

*The data in 2018 were obtained from the public data base (see the text).

The significant results were analyzed further by the residual analysis.

△ and ▼ denote more and fewer significantly in the comparison by the residual analysis, respectively.

(-) denote no significant differences in the comparison made using the residual analysis.

NA: not applicable. NS: no significant differences.

[1] 276 samples from 273 participants. Three participants submitted two samples separately on the different days.

Examination Institute. In terms of sex, all three groups were predominantly female and there were no significant differences between the groups. In terms of age groups, similar proportions were observed in three age groups. The CPP group had a significantly higher proportion of examinees in their 40s and a significantly lower proportion in their 60s than the hospital/clinic institution group. The CPP group had a significantly higher proportion of patients in their 40s and 50s and a significantly lower proportion of patients in their 60s than the hospital/clinic and the health examination institute groups. Compared between the hospital/clinic institution and health examination institute groups, the hospital/clinic institution group had a significantly lower proportion of patients in their 40s and 50s and a significantly higher proportion of patients in their 60s.

There were no significant differences between the three groups in terms of FIT experience. Although no data were reported from hospitals, clinics, and health examination institutions, the community pharmacies used the 2-day FIT method 100% of the time, and had a turn-around time of 1 d. In total, 270 samples were simultaneously submitted by community pharmacies, while 3 samples were submitted separately on different days.

## Results of detailed examinations

As shown in Table 3, the CPP group did not differ significantly from the hospital/clinic or health examination institution groups in terms of sex and number of detailed examination findings. In contrast, the proportion of patients who underwent

**Table 3. Findings on the second examination of the applicants showing positive results of the fecal immunochemical test.**

| | Procedures | | | | | |
| --- | --- | --- | --- | --- | --- | --- |
| | CPP (A) | *Hospital/clinic (B) | *Health examination institution (C) | A vs B | A vs C | B vs C |
| Total number, n | 15 | 126 | 423 | | | |
| Sex | | | | NS | NS | NS |
| male, n (%) | 8 (53.3) | 52 (41.3) | 150 (35.5) | | | |
| female, n (%) | 7 (46.6) | 74 (58.7) | 273 (64.5) | | | |
| Second examination performed | | | | p<0.05 | p<0.05 | NS |
| | | | | A B | A C | |
| Yes, n (%) | 7 (46.7) | 110 (87.3) | 351 (83.0) | ▼ △ | ▼ △ | |
| No, n (%) | 8 (53.3) | 16 (12.7) | 72 (17.0) | △ ▼ | △ ▼ | |
| Yes, number in the 40s, 50s, and 60s | 2, 1, 4 NS# | 13, 16, 81NS# | 59, 43, 249 p<0.01# | | | |
| No, number in the 40s, 50s, and 60s | 2, 2, 4 | 1, 1, 14 | 23, 16, 33 | | | |
| Findings | | | | NS# | NS# | p<0.01 |
| | | | | | | B C |
| Malignancy, n (%) | 0 (0.0) | 8 (7.3) | 16 (4.6) | | | - - |
| Adenoma, n (%) | 4 (57.1) | 58 (52.7) | 111 (31.6) | | | △ ▼ |
| Other findings, n (%) | 1 (14.3) | 20 (18.2) | 119 (33.9) | | | ▼ △ |
| NP, n (%) | 2 (28.6) | 24 (21.8) | 105 (29.9) | | | - - |

CPP: community pharmacy pharmacist. *The data in 2018 were obtained from the public data base (see the text).

The statistical analysis was performed using the Chi square test except findings.

#The findings results were analyzed using Fisher's exact test due to the small size of the sample.

The significant results were analyzed further using the residual analysis.

△ and ▼ denote more and fewer significantly in the comparison using the residual analysis, respectively.

(-) denote no significant differences in the comparison using the residual analysis.

NP: no particular findings. NS: no significant differences

a detailed examination was significantly lower in the CPP group than in the hospital/clinic and health examination institution groups. There was no significant difference between the hospital/clinic institution and health examination institution groups in terms of sex or rate of visits for detailed examinations. The proportions of patients who underwent a detailed examination in their 40s, 50s, and 60s were not significantly different between the CPP and hospital/clinic institution groups. In contrast, the proportion of patients who underwent a detailed examination in their 40s, 50s, and 60s were significantly different in the health examination institution group. The hospital/clinic institution group had significantly more adenomas and lesions other than adenomas and cancers compared to the health examination institution group.

## Discussion

More than one-third of the FIT for CPP participants included in the study were not taking any medication at the pharmacy where they were evaluated. Compared with conventional CRC screening procedures, a higher percentage of the younger adults were screened, however the rate of detailed examination was lower. The 2-day method was used 100% of the time. In addition, the rate of pathological findings on the detailed examination in the CPP-mediated CRC screening group was similar to that in the hospital/clinic and health examination groups. These results suggest that the CPP-mediated screening could be used as a novel screening procedure for CRC.

In the current study, approximately 10% of city pharmacies participated in the study. The participating pharmacies had 1 to 14.4 working pharmacists, with the number of working pharmacists positively correlating with the number of kits distributed (r = 0.454, $p < 0.01$). However, there were no significant differences in the number of kits distributed by pharmacies with ≤2 versus >2 working pharmacists, suggesting that this testing support could be implemented even through pharmacies with a small number of pharmacists on duty. This suggests that this initiative is feasible in other regions of Japan, where there are many pharmacies with a small number of pharmacists on duty [22].

Those who participated in the study were more commonly enrolled in the Japan Health Insurance Association, which covers people working for small- and medium-sized companies than in the National Health Insurance, which covers people who are self-employed, or the Union Health Insurance, which covers people working for large-sized companies [23]. In Japan, municipalities are responsible for cancer screening, including CRC screening [6]. However, the percentage of working individuals who undergo CRC screening has recently increased [7]. However, the completion rate depends on the company size. Large company workers have the opportunity to receive CRC screening tests as part of their medical examinations, while small- and medium-sized company workers often do not have CRC screening tests as part of their medical examinations [24]. In Kagoshima, a regional city, a high percentage of small- and medium-sized companies are expected to have low CRC screening rates. Therefore, a CPP-mediated CRC test supporting system may be necessary for local cities with people working for many small- and medium-sized companies.

In the current study, one out of ten individuals who received the FIT kit performed FIT (Fig 1). The implementation rates in CPP-mediated procedure were low compared to that in Spain, where 40% of individuals receiving an invitation letter finally performed FIT [19]. This was attributed to the financial burden of implementation on participants. The rates reported in Spain were based on the number of people tested per number of recommendations in letters, while those in the current study were based on the number of people tested per number of kits submitted, making comparison between these two studies difficult. However, the current study may not have been more convenient due to the limited number of participating pharmacies, and the costs associated with the test, resulting in a lower implementation rate.

Due to prospects of implementing this testing procedure in Japan, the financial burden was set at 1,100 JPY in the current study, considering the costs of medical institution-mediated procedures (1,100 JPY) and health examination institution-mediated procedures (600 JPY) as self-payments, both of which are already in place in Kagoshima. In addition, the two procedures conducted through the Health Promotion Act are financially supported by the prefectural and/or city office. In contrast, the CPP-mediated procedure costed the test kit (165 JPY) and implementation of the test (435 JPY), which generated a profit of 500 JPY which was used to fund the research study. A profit of 500 JPY will be supplied to pharmacies for future implementation. It should be considered that a procedure can be implemented and sustained in society without government support, compared to the financial support of government-led procedures used in other regions [17,18]. Besides the financial benefits for pharmacies, this implementation was an opportunity to put into practice the social contribution of supporting the health of the local population and workplace employees around the pharmacy.

Regarding the age of the participant, the CPP-mediated procedures had a small bias in the age categories of 40s, 50s, and 60s. However, the other two groups' participants were mostly in their 60s, suggesting that it may be possible to target a relatively younger generation through CPP-mediated procedures. A relatively large number of people were enrolled in the Association Health Insurance Plan. Therefore, employees of small and medium-sized enterprises who had not yet undergone CRC screening were able to take the test. Women of working age who visited the pharmacy with their children or parents may have also been screened. These suggest that if the pharmacies were in the community and provided an environment conducive to screening, then the applicants in the CPP-mediated procedure were relatively young.

Moreover, it has been reported that the test itself, even if it is a 2-day method, is required to increase sensitivity compared with the one-day method [25]. The fact that the 2-day method was used 100% of the time in the current study may have increased the positivity rate [26]. Therefore, future studies with larger sample sizes are warranted. The test results were returned within two days. These results were similar to those obtained in Spain [19]. The fact that the results

returned immediately after the test indicates a shorter period of anxiety while waiting for the test results, suggesting this as an advantage of conducting CRC tests in community pharmacies.

FITs performed at high ambient temperatures have low positivity rates and sensitivities [27]. For testing, it is believed that the samples were properly managed, as was the drug management in the community pharmacies. It has been noted that the use of kits purchased over the mail-based self-sampling resulted in poor handling, including the transportation condition of the collected stools [28]. Although mail-based self-sampling is reported to be cost-effective [29], it may be associated with problems with the storage and transportation of samples and is not a recommended procedure due to its high false negative rates [28]. In Japan, the cost of mail-based self-sampling when purchased via the Internet is also high.

Contrary to expectations, the rate of undergoing a detailed examination was lower for CPP-mediated procedure than for the other two. This rate was expected to be higher in CPP-mediated procedure because the pharmacists can provide individualized information on medical institutions that perform detailed examinations. The people who received the detailed examinations in the CPP-mediated procedure were relatively young (40–59 years: 46.7%, Table 3). Fewer people in their 40s and 50s received a detailed examination in the health examination institution-mediated procedure than those in their 60s. The relatively high proportion of young people among those with a positive FIT may be one of the reasons for the low rate of detailed examinations in the CPP-mediated procedure.

Those who were originally less interested in undergoing medical check-ups might have tested positive. Even if the results were positive, this may not have influenced them to take the next step. In Japan, a high primary screening uptake rate has been reported for CRC conducted in the workplace [30,31]. However, the rate of receiving detailed examinations is low (3040%). Our current results are similar to those of screenings conducted in the workplace. To increase the CRC screening rate, including the rate of receiving a detailed examination, enhanced programs such as educational booklets and telephone calls may be required [32]. However, the prevalence of pathological findings was not significantly different between the other two groups (Table 3), suggesting that CPP-mediated CRC screening is meaningful.

Furthermore, a unique Japanese system of school pharmacist activities can be used to increase the implementation rate. Health information was provided to school children [33,34]. By using this system to provide information about CRC testing from children to their parents, it is expected that the number of participants who will receive the kit at a nearby pharmacy and undergo the test will increase.

This study had several limitations. First, the kits used in the study were distributed for one month in March. The short distribution period could have affected the results. However, the procedure was conducted at the end of the fiscal year (March), in anticipation of those who had not yet been screened, as regular cancer screening begins in April in Japan. In the future, it could be conducted throughout the year, with the importance of serving as a contact point for those who have not yet undergone regular cancer screening, and especially with a reinforcement month at the end of the fiscal year. Second, this study reported the cost of implementing FIT. Verification of cost-effectiveness was required in several places. In this study, the verification was done with the fewest variables. More valuable evidence was needed to determine whether the CPP-mediated procedure can be sustained in real world settings [35]. Finally, the number of samples distributed was uneven among pharmacies. The location of the pharmacies, the medical specialties of nearby medical facilities, and the enthusiasm of the participating pharmacists may have influenced their distribution. Although the pharmacists volunteered to participate in this implementation, there was variability in how they communicated the procedure. This variability may have influenced the actual CRC screening rate.

## Conclusions

The introduction of a screening test for CRC prevention through the participation of CPPs has been discussed. Compared to conventional procedures, a higher percentage of the younger adult population took the test in community pharmacies, suggesting that the procedure could be used as a novel screening procedure for CRC. It should be considered that a procedure can be implemented and sustained in society without government support. The uptake rate in the current study

was low. However, as the procedure requires the recipient to bear the cost, increasing the uptake rate could help to financially sustain this procedure. In addition, motivation for the pharmacists themselves to engage in the activity is likely to exist, as it increases the social value of CPPs. This model should be widely known, even among people with lower health awareness. Posters and brochures about this CPP-mediated procedure should be prepared for visitors to pharmacies and public events.

## Supporting information

**S1 Appendix. Fig 1 of a general briefing.**
(TIF)

**S2 Appendix. Fig 2 of a general briefing.**
(TIF)

**S3 Appendix. Fig 3 of a general briefing.**
(TIF)

**S4 Appendix. Fig 4 of a general briefing.**
(TIF)

**S5 Appendix. Fig 5 of a general briefing.**
(TIF)

**S6 Appendix. Fig 6 of a general briefing.**
(TIF)

**S7 Appendix. Fig 7 of a general briefing.**
(TIF)

**S8 Appendix. Fig 8 of a general briefing.**
(TIF)

**S9Appendix. Fig 9 of a general briefing.**
(TIF)

## Acknowledgments

We thank editage (https://www.editage.jp) for editing the manuscript. This manuscript, written in Japanese, was partly translated using Deep L AI software.

## Author contributions

**Conceptualization:** Masahisa Horiuchi, Yusuke Jin'o.

**Data curation:** Yusuke Jin'o, Miharu Ushikai.

**Formal analysis:** Masahisa Horiuchi, Yusuke Jin'o, Miharu Ushikai.

**Funding acquisition:** Masahisa Horiuchi, Miharu Ushikai.

**Investigation:** Masahisa Horiuchi, Yusuke Jin'o.

**Methodology:** Masahisa Horiuchi, Yusuke Jin'o, Miharu Ushikai.

**Project administration:** Masahisa Horiuchi.

**Validation:** Masahisa Horiuchi, Yusuke Jin'o, Miharu Ushikai.

**Visualization:** Yusuke Jin'o, Miharu Ushikai.

**Writing – original draft:** Yusuke Jin'o.

**Writing – review & editing:** Masahisa Horiuchi, Miharu Ushikai, Yuga Komaki, Koichi Masuda.

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
