## [Decision Letter · Decision Letter 0]

8 Dec 2024

PONE-D-24-43344Involvement of the community pharmacy pharmacists in fecal immunochemical test screening without government supports in JapanPLOS ONE

Dear Dr. Horiuchi,

Thank you for submitting your manuscript to PLOS ONE. After careful consideration, we feel that it has merit but does not fully meet PLOS ONE’s publication criteria as it currently stands. Therefore, we invite you to submit a revised version of the manuscript that addresses the points raised during the review process.

We look forward to receiving your revised manuscript.

Kind regards,

Sameen Abbas

Academic Editor

PLOS ONE

Journal Requirements:

Reviewers' comments:

Reviewer's Responses to Questions

**Comments to the Author**

1. Is the manuscript technically sound, and do the data support the conclusions?

Reviewer #1: Partly

Reviewer #2: Yes

2. Has the statistical analysis been performed appropriately and rigorously? 

Reviewer #1: No

Reviewer #2: Yes

3. Have the authors made all data underlying the findings in their manuscript fully available?

Reviewer #1: Yes

Reviewer #2: Yes

4. Is the manuscript presented in an intelligible fashion and written in standard English?

Reviewer #1: Yes

Reviewer #2: Yes

5. Review Comments to the Author

Reviewer #1: 1. Introduction needs to be reorganized (paragraph 1; burned and incidence including Japa, 2: uptake and screening, 3: health promotion, 4; the Community pharmacy pharmacists and study aim), need to be consistence, if we say uptake should be uptake throughout, if screening rate should have to the same throughout, there is a need for rephrasing and rearrangement of the paragraphs.

2. Introduction 49 to 51; this has to be part of second paragraph

3. Introduction line 52; this sound to be part of paragraph one which speak on burden and has to include the incidence as well.

4. Introduction line 66 to 67; is not clear what is six-years system mean and why is so important for this study.

5. Methodology Participants: line 81 to 90; you have to explain the characteristics of participants/ or participating pharmacies, and including their exclusion criteria but I see also you included the part of the results in this section.

6. Methodology is missing the study design part, is it cross-sectional, survey, or randomized study???

7. Methodology Examinations line 92; this has to be rephrased to study procedure

8. Methodology line 93 to 95; the authors need to mention the sensitivity cut point you are referring to is it 98%, 35% etc and explain the usually CRC screening Method used in Japan. And you need to start this procedure part by explaining what was done, how, when and where, and you mention the kits used for this and its sensitivity.

9. Methodology data line 100 this has to be data collection method.

10. Methodology data line 106 to 109; the authors need to stick to CRC and not COVID-19.

11. Result Correlation line 131; why did we do the correlation; do we have correlation procedure in analysis like Kappa.

12. Result Table 1: need to be the table for participants characteristics, age., sex, how many tested (tested negative and tested positive), insurance (mention only the insurance with significance number, and the other with lower number should be grouped under one part mentioned as other insurance). The one you provide fits to be part of supplementary tables.

13. Result line 140 to 143; this is parts for participant in methodology and not results; Authors need to remove the results in this section and include it on respective section

14. Result line 156 to 158; it has not explained well in introduction even on the methodology and statistical part if it is needed it needs to be reflected on the study aim on introduction and well as in analysis procedure.

15. Discussion of this study is not clear and straightforward, the authors explained all results as in results section, some are not in the scope of this study, in general needs to be focused.

16. Discussion; the first paragraph needs to have your 3 or 4 important findings according to your aim. Thereafter, the following paragraphs has to contain the detailed information of each mentioned important findings from 1st paragraph, you compare and contrast against evidence from previously.

17. Discussion line 219 to 223; no need for this information in this part of discussion advised to be deleted.

18. Discussion Line 245 to 247; need to be consistence on wording choose either pharmacist-mediated CRT test or community pharmacy pharmacists, check also other part of manuscript where you mixed up these words.

19. Discussion line 249; no need to refer figure, you already mentioned in results

20. Discussion line 248 to264; this paragraph has the important finding according to the study aim the uptake following the intervention, you need to explain in detains on the uptake, comparing with other findings from other studies, and give the impression for it. The cost and profit need to be in a separate paragraph, and include in the study procedure the information on how you assessed the cost and profit.

21. Discussion line 265 to 275; need to be a separate paragraph, no need to refer table at this point already is in the result. Rephrase and mention only the important finding of which I see the uptake of community pharmacy pharmacist test CRC test by young age. Remove the other part which has no information on the uptake and the paragraph should be before the const information. And discuss in details on your result about young age are these also seen to other studies from different area or not, and if possible, why.

22. Discussion line 275 to 294; The positivity rate increase with age is important information but has not discussed well and is not clear if you’re reporting this as a limitation or you are reporting the new findings, need to separate this paragraph part of it is not scope of this study.

23. Discussion line 295 to 296; is a good finding but what are the other two you mean needs more discussion on this finding.

24. Discussion line 319 to 320; reporting the cost why it is a limitation to this study?

Reviewer #2: Thank you for the manuscript. I have a few questions regarding its content:

Abstract:

1. Could key statistical findings be added to better highlight the study’s impact?

2. How does the community pharmacy approach specifically address gaps in CRC screening participation compared to other methods?

Methods:

3. Was there variability in how pharmacists communicated the procedure, and could this variability have influenced participant uptake?

4. Were standardized protocols in place to ensure proper storage and transport of the returned FIT kits?

Results:

5. Why was the uptake rate lower compared to international benchmarks? Could cultural or logistical factors have played a role?

6. What could explain the significantly lower rate of undergoing detailed examinations in the pharmacy group?

Discussion:

7. How might this model help to address colorectal cancer screening gaps in rural or underserved populations?

8. What strategies could be implemented to improve participation among men and individuals with lower health awareness?

Limitations:

9. Could the short distribution period have influenced the results, and how could this issue be mitigated in future studies?

The manuscript is generally well-written, but there are areas where the language could be improved for clarity and professionalism.

6. PLOS authors have the option to publish the peer review history of their article (what does this mean? ). If published, this will include your full peer review and any attached files.

**Do you want your identity to be public for this peer review?** For information about this choice, including consent withdrawal, please see our Privacy Policy .

Reviewer #1: No

Reviewer #2: No

---

## [Author Response · Author response to Decision Letter 1]

10 Jan 2025

Sameen Abbas, Ph.D.

Academic Editor

PLOS ONE

Dear Prof. Abbas,

We are pleased to learn that our manuscript (PONE-D-24-43344, Involvement of the

community pharmacy pharmacists in fecal immunochemical screening without government

supports in Japan) can be considered for publication in your journal.

First of all, we explained the correction to the Table 2. We have noticed the error in in Table 2 regarding the applicants aged 50-59 in the hospital/clinic-mediated procedure. According to the correction, we have corrected the statistical results, which are marked in red. The percentage of positive results in the hospital/clinic-mediated procedure was changed from 5.8 to 5.7. In addition, we have added the data on the number of age groups (40s, 50s, and 60s) in the hospital/clinic- and health examination institution-mediated procedures in Table 3 to respond to the reviewers’ comments. The correction and addition would be helpful for a better understanding of the presentation of our manuscript.

We have responded to the reviewers’ comments point by point as follows. We have highlighted in red the sentences that have been added to the text.

Reviewer #1:

1. Introduction needs to be reorganized (paragraph 1; burned and incidence including Japan, 2: uptake and screening, 3: health promotion, 4; the Community pharmacy pharmacists and study aim), need to be consistence, if we say uptake should be uptake throughout, if screening rate should have to the same throughout, there is a need for rephrasing and rearrangement of the paragraphs.

We agreed with the comments. We have reorganized the introduction section as suggested by the queries 2-3 as below.

The screening rate is used consistently.

2. Introduction 49 to 51; this has to be part of second paragraph

We have moved this part to the second paragraph.

3. Introduction line 52; this sound to be part of paragraph one which speak on burden and has to include the incidence as well.

We have moved this part to the first paragraph.

4. Introduction line 66 to 67; is not clear what is six-years system mean and why is so important for this study.

The change from a four-year system to a six-year system has increased the increased content on prevention and health in addition to clinical training. Therefore, the change to the six-year system is important for this study. Accordingly, we changed the sentence (P4 L70-72 in the Manuscript).

However, pharmacists in Japan are undergoing a shift to a six-year system at university and the expansion of the educational content related to prevention and health in addition to clinical training [16].

5. Methodology Participants: line 81 to 90; you have to explain the characteristics of participants/ or participating pharmacies, and including their exclusion criteria but I see also you included the part of the results in this section.

We have removed the results part of this section. We have added the information to the sentence (P5 L84-85 in the Manuscript).

Thirty-seven community pharmacies in Kagoshima City voluntarily participated in the study (Table 1).

6. Methodology is missing the study design part, is it cross-sectional, survey, or randomized study???

We have added the part in this section as Study design and participants. We added the sentence (P5 L84 in the Manuscript).

This study was conducted as a survey.

7. Methodology Examinations line 92; this has to be rephrased to study procedure

We have combined the sections (Participants and Examinations) and rephrased them as Study design and procedure.

8. Methodology line 93 to 95; the authors need to mention the sensitivity cut point you are referring to is it 98%, 35% etc and explain the usually CRC screening Method used in Japan. And you need to start this procedure part by explaining what was done, how, when and where, and you mention the kits used for this and its sensitivity.

The 2-day method used in the study is the usual CRC screening method in Japan.

We modified the sentence as follows (P5 L94-96 in the Manuscript).

Considering the sensitivity of the test, a 2-day method was used, which is the usual method for CRC screening in Japan [20].

As suggested, we explained what was done, how, when, and where in the study design and procedure section (P5 L86-93 in the Manuscript).

Kits were distributed by participating pharmacies between March and the first week of April 2023 (Fig 1). Stool samples were collected at home, brought to a community pharmacy, and sent to the laboratory (Clinical Pathology Laboratory Co. Ltd., Kagoshima, Japan). The collection period was from March to August 31, 2023. The results were reported to the community pharmacist in charge and then to the participants. For patients who tested positive, explanations were provided about the medical institutions that performed detailed examinations.

We have added the sentences (P5-6 L96-98 in the Manuscript).

According to the manufacturer, the cut-off value is 100 ng Hb/ml. Therefore, the sensitivity of the 2-day method for CRC is 100.0% (74.1-100.0% as 95% confidential intervals) [20].

9. Methodology data line 100 this has to be data collection method.

We have rephrased it as suggested.

10. Methodology data line 106 to 109; the authors need to stick to CRC and not COVID-19.

We have explained the sentences. These sentences are the reason for selecting the data.

We have added the sentence and removed the sentence for the redundancy (P7 L120-122 in the Manuscript).

The time of the data was chosen because the COVID-19 pandemic could affect the screening rate of general health examinations, including CRC screening.

11. Result Correlation line 131; why did we do the correlation; do we have correlation procedure in analysis like Kappa.

As we understand it, the kappa statistic is used to assess the reproducibility using the same samples. Since the participants in each pharmacy were different, we analyzed the correlation between the number of pharmacists in the pharmacy and the number of kits distributed in the pharmacy.

12. Result Table 1: need to be the table for participants characteristics, age., sex, how many tested (tested negative and tested positive), insurance (mention only the insurance with significance number, and the other with lower number should be grouped under one part mentioned as other insurance). The one you provide fits to be part of supplementary tables.

We are sorry that we did not receive information about the participants who did not take FIT. Therefore, we cannot prepare the table as suggested. We have corrected the ethical sentence (P8 L135-136 in the Manuscript).

Informed consent was obtained from all participants with FIT examination before they completed the questionnaire.

13. Result line 140 to 143; this is parts for participant in methodology and not results; Authors need to remove the results in this section and include it on respective section

We have moved it to the methodology section.

14. Result line 156 to 158; it has not explained well in introduction even on the methodology and statistical part if it is needed it needs to be reflected on the study aim on introduction and well as in analysis procedure.

We have added the explanations in the Introduction section (P3 L57-58 in the Manuscript).

In Japan, there are two FIT procedures that are provided by hospitals/clinics and health examination institutes.

15. Discussion of this study is not clear and straightforward, the authors explained all results as in results section, some are not in the scope of this study, in general needs to be focused.

We agree with the proposal. We have attempted to rewrite the discussion section as indicated below.

16. Discussion; the first paragraph needs to have your 3 or 4 important findings according to your aim. Thereafter, the following paragraphs has to contain the detailed information of each mentioned important findings from 1st paragraph, you compare and contrast against evidence from previously.

We described the key points of the study (P18 L229-236 in the Manuscript).

CPP-mediated CRC screening was conducted in Japan. More than one-third of the FIT participants with FIT were not taking any medication at the pharmacy where they were tested. Compared with conventional CRC screening procedures, a higher percentage of the younger generation was screened, but the rate of detailed examination was lower. The 2-day method was used 100% of the time. In addition, the rate of pathological findings on the detailed examination in the CPP-mediated CRC screening was similar to that in the hospital/clinic and health examination-mediated procedures. These results suggest that the CPP-mediated procedure could be used as a novel screening procedure for CRC.

17. Discussion line 219 to 223; no need for this information in this part of discussion advised to be deleted.

We have removed the sentences.

18. Discussion Line 245 to 247; need to be consistence on wording choose either pharmacist-mediated CRT test or community pharmacy pharmacists, check also other part of manuscript where you mixed up these words.

We have used the term, community pharmacy pharmacists as CPPs, consistently throughout the text.

19. Discussion line 249; no need to refer figure, you already mentioned in results

We have deleted the information.

20. Discussion line 248 to264; this paragraph has the important finding according to the study aim the uptake following the intervention, you need to explain in detains on the uptake, comparing with other findings from other studies, and give the impression for it. The cost and profit need to be in a separate paragraph, and include in the study procedure the information on how you assessed the cost and profit.

We appreciated the suggestion. We have detailed the comparison of the implementation rate in Spain (P19-20 L262-267 in the Manuscript).

Spain was based on the number of people tested per number of recommendations in letters, while the current study was based on the number of people tested per number of kits distributed, so there is no easy comparison, but the current study may not have been more convenient due to the limited number of participating pharmacies, and the costs associated with the test were charged, resulting in a lower implementation rate.

We separated the paragraph in the discussion section, and added the information on the cost of the procedures (hospital/clinic-mediated procedure and health examination institution-mediated procedure) in the study procedure section (P6 L103-105 in the Manuscript).

According to the local government rule, the other two procedures (hospital/clinic-mediated procedure and health examination institution-mediated procedure) are performed by the cost of self-payment, 600 JPY and 1100 JPY, respectively.

21. Discussion line 265 to 275; need to be a separate paragraph, no need to refer table at this point already is in the result. Rephrase and mention only the important finding of which I see the uptake of community pharmacy pharmacist test CRC test by young age. Remove the other part which has no information on the uptake and the paragraph should be before the const information. And discuss in details on your result about young age are these also seen to other studies from different area or not, and if possible, why.

We have removed the part that does not provide information on the screening rate as suggested. We have separated the paragraph related to the sensitivity of the test.

We discussed in detail the results of the higher screening rate at younger ages in the CPP-mediated procedure. We have added the following sentences (P21 L285-291 in the Manuscript).

A relatively large number of people were enrolled in the Association Health Insurance Plan. Therefore, employees of small and medium-sized enterprises who had not yet undergone CRC screening were able to take the test. Women of working age who visited the pharmacy with their children or parents may have been screened. These suggest that the pharmacies were in the community and provided an environment conducive to screening, then the applicants in the CPP-mediated procedure were relatively young.

22. Discussion line 275 to 294; The positivity rate increase with age is important information but has not discussed well and is not clear if you’re reporting this as a limitation or you are reporting the new findings, need to separate this paragraph part of it is not scope of this study.

We agree with the suggestion and have removed this part.

23. Discussion line 295 to 296; is a good finding but what are the other two you mean needs more discussion on this finding.

We analyzed the people who received the detailed examinations in the hospital/clinic- procedure and in the health examination institution-mediated procedures. Fewer people in their 40s and 50s received a detailed examination in the health examination institution-mediated procedure compared with people in their 60s. The relatively high proportion of young people among those with a positive FIT is one of the reasons for the low rate of detailed examinations in the CPP- mediated procedure.

We have added the data to Table 3, and we added the following sentences in the Discussion section (P22 L311-316 in the Manuscript).

The people who received the detailed examinations in the CPP-mediated procedure were relatively young (40-59 years: 46.7%, Table 3). Fewer people in their 40s and 50s than in their 60s received a detailed examination in the health examination institution mediated procedure. The relatively high proportion of young people among those with a positive FIT may be one of the reasons for the low rate of detailed examinations in the CPP-mediated procedure.

24. Discussion line 319 to 320; reporting the cost why it is a limitation to this study?

The cost verification in this study was done with the fewest variables. We thought that more careful verification was needed, so the cost calculation is raised as a limitation. We have modified the sentences according to the concept (P23 L337-340 in the Manuscript).

Second, this study reported the cost of implementing FIT. Verification of cost-effectiveness was required in several places. In this study, the verification was done with the fewest variables. More valuable evidence was needed to determine whether the CPP-mediated procedure can take root in society [36].

Reviewer #2: Thank you for the manuscript. I have a few questions regarding its content:

Abstract:

1. Could key statistical findings be added to better highlight the study’s impact?

We appreciate the suggestion. We have added the main statistical results in the section.　

We have added the following sentences in the Abstract section as follows (P2 L31-36 in the Manuscript).

A significantly higher percentage of people in their 40s underwent the test in the CPP-mediated procedures (35.2% vs. 14.3% in the hospital/clinic, and 21.8% in the health examination institution, respectively, p < 0.01). The percentage of participants who underwent a detailed examination was significantly lower than in the other two groups (46.7% vs. 87.3% in the hospital/clinic, and 83.0% in the health examination institution, respectively, p < 0.05).

2. How does the community pharmacy approach specifically address gaps in CRC screening participation compared to other methods?

We appreciate the suggestion. We have added the sentences (P3 L40-42 in the Manuscript).

In particular, the CPP-mediated procedure may address gaps by increasing the number of young people and people with low opportunity to receive CRC screening.

Methods:

3. Was there variability in how pharmacists communicated the procedure, and could this variability have influenced participant uptake?

Yes, there was. The degree to which pharmacists communicated the procedure may have influenced participants' uptake. We have added these sentences to the study limitation in the Discussion section (P24 L343-346 in the Manuscript).

Although the pharmacists volunteered to participate in this implementation, there was variability in how they communicated the procedure. This variability may have influenced the actual CRC screening rate.

4. Were standardized protocols in place to ensure proper storage and transport of

---

## [Decision Letter · Decision Letter 1]

16 Mar 2025

PONE-D-24-43344R1Involvement of the community pharmacy pharmacists in fecal immunochemical test screening without government supports in JapanPLOS ONE

Dear Dr. Horiuchi,

Thank you for submitting your manuscript to PLOS ONE. After careful consideration, we feel that it has merit but does not fully meet PLOS ONE’s publication criteria as it currently stands. Therefore, we invite you to submit a revised version of the manuscript that addresses the points raised during the review process.

We look forward to receiving your revised manuscript.

Kind regards,

Sameen Abbas

Academic Editor

PLOS ONE

Journal Requirements:

**Additional Editor Comments:**

Revise per reviewer's comments

Reviewers' comments:

Reviewer's Responses to Questions

**Comments to the Author**

1. If the authors have adequately addressed your comments raised in a previous round of review and you feel that this manuscript is now acceptable for publication, you may indicate that here to bypass the “Comments to the Author” section, enter your conflict of interest statement in the “Confidential to Editor” section, and submit your "Accept" recommendation.

Reviewer #1: All comments have been addressed

Reviewer #3: (No Response)

2. Is the manuscript technically sound, and do the data support the conclusions?

Reviewer #1: Yes

Reviewer #3: Partly

3. Has the statistical analysis been performed appropriately and rigorously? 

Reviewer #1: Yes

Reviewer #3: Yes

4. Have the authors made all data underlying the findings in their manuscript fully available?

Reviewer #1: Yes

Reviewer #3: Yes

5. Is the manuscript presented in an intelligible fashion and written in standard English?

Reviewer #1: Yes

Reviewer #3: No

6. Review Comments to the Author

Reviewer #1: Thank you for revising this manuscript now good and great for publication

one minor comment other wise good.

1. Methodology: Minor additional suggestion on ethics. Would suggest to remove line 135 to 136: Informed consent was obtained from all participants with FIT examination before they completed the questionnaire; sounds like repetition and remain with line 137 which cover both FIT examination and enrollment.

Reviewer #3: The grammar needs to be improved throughout the manuscript. The language is not good. Also, the main message of the paper is not clear. What's the main takeaway of the paper that the reader should have. I was confused when I was reading the paper about the intent of the authors. Are they describing how community pharmacies don't have enough uptake? They conclude that the procedure can be sustained without financial incentives, but where do authors state this in the paper?

7. PLOS authors have the option to publish the peer review history of their article (what does this mean? ). If published, this will include your full peer review and any attached files.

**Do you want your identity to be public for this peer review?** For information about this choice, including consent withdrawal, please see our Privacy Policy .

Reviewer #1: No

Reviewer #3: No

---

## [Author Response · Author response to Decision Letter 2]

25 Mar 2025

Sameen Abbas, Ph.D.

Academic Editor

PLOS ONE

Dear Prof. Abbas,

We are pleased to learn that our manuscript (PONE-D-24-43344_R2, Involvement of community pharmacy pharmacists in fecal immunochemical screening without government support in Japan) can be considered for publication in your journal.

We have responded to the reviewers’ comments point by point below. We have highlighted the sentences that have been added to the text in red.

Reviewer #1:

1. Methodology: Minor additional suggestion on ethics. Would suggest to remove line 135 to 136: Informed consent was obtained from all participants with FIT examination before they completed the questionnaire; sounds like repetition and remain with line 137 which cover both FIT examination and enrollment.

Thank you for your suggestion. As suggested, we have deleted the sentence.

Reviewer #3:

1. The grammar needs to be improved throughout the manuscript. The language is not good.

We have asked a native speaker, through an editing company, to proofread the manuscript. We hope this has improved the language and grammar of the manuscript. We have also included the English editing certificate for your reference.

2. The main message of the paper is not clear. What's the main takeaway of the paper that the reader should have.

We appreciate your question. The main message of the paper is the possible contribution that CPPs can make towards CRC prevention through FIT. This has been added in Conclusions section.

3. I was confused when I was reading the paper about the intent of the authors. Are they describing how community pharmacies don't have enough uptake? They conclude that the procedure can be sustained without financial incentives, but where do authors state this in the paper?

We appreciate your comment. Accordingly, we have added the following sentences to the Conclusions section to present the relevant information (L349-352 in the manuscript):

The uptake rate in the current study was low. However, as the procedure requires the recipient to bear the cost, increasing the uptake rate could help to financially sustain this procedure. In addition, motivation for the pharmacists themselves to engage in the activity is likely to exist, as it increases the social value of CPPs.

---

## [Editor Report · Decision Letter 2]

30 Mar 2025

Involvement of community pharmacy pharmacists in fecal immunochemical test screening without government support in Japan

PONE-D-24-43344R2

Dear Dr. Masahisa Horiuchi,

We’re pleased to inform you that your manuscript has been judged scientifically suitable for publication and will be formally accepted for publication once it meets all outstanding technical requirements.

Kind regards,

Sameen Abbas

Academic Editor

PLOS ONE

---

## [Editor Report · Acceptance letter]

PONE-D-24-43344R2

PLOS ONE

Dear Dr. Horiuchi,

I'm pleased to inform you that your manuscript has been deemed suitable for publication in PLOS ONE. Congratulations! Your manuscript is now being handed over to our production team.

Kind regards,

on behalf of

Dr. Sameen Abbas

Academic Editor

PLOS ONE